# Dysfunction of peripheral somatic and autonomic nervous system in patients with severe forms of Crohn's disease on biological therapy with TNFα inhibitors–A single center study

Martin Wasserbauer[1], Sarka Mala[1], Katerina Stechova[1], Stepan Hlava[1]*,
Pavlina Cernikova[1], Jan Stovicek[1], Jiri Drabek[1], Jan Broz[1], Dita Pichlerova[1],
Barbora Kucerova[2], Petra Liskova[3], Jan Kral[4], Lucia Bartuskova[5], Radan Keil[1]

1 Department of Internal Medicine, 2nd Faculty of Medicine, Charles University in Prague and Motol University Hospital, Prague, Czech Republic, 2 Department of Pediatric Surgery, 2nd Faculty of Medicine Charles University in Prague and University Hospital Motol, Prague, Czech Republic, 3 Department of Neurology, 2nd Faculty of Medicine, Charles University and University Hospital Motol, Prague, Czech Republic, 4 Department of Gastroenterology and Hepatology, Institute for Clinical and Experimental Medicine, Prague, Czech Republic, 5 Department of Economic and Social Policy, University of Economics, Prague, Czech Republic

* stepan.hlava@fnmotol.cz

## Abstract

### Objective

Crohn's disease (CD) can be associated with a wide range of extraintestinal manifestations (EIMs), including neurological ones. Published studies differ in their conclusions about the epidemiology and etiopathogenesis of neurological EIMs. The aims of this study were to demonstrate the presence and find risk factors of peripheral (somatic and autonomic) neuropathy patients with severe CD on anti-TNFα biological therapy.

### Material and methods

A clinical examination focusing on detection of peripheral sensor-motor nervous dysfunction (including Sudoscan) and examination of autonomic nervous system dysfunction (using Ewing´s battery tests and spectral analysis) together with laboratory tests and collection of demographic data followed by administration of questionnaires were performed on a total of 30 neurologically asymptomatic outpatients with severe CD on anti-TNFα biological therapy.

### Results

Peripheral sensor-motor nervous function via clinical neurological examination was pathological in 36.7% and Sudoscan in 33.3% of cases. Statistically significant associations between vibration perception test and age, CD and biological therapy duration, body mass index and Crohn's Disease Activity Index were proved while statistically significant

**Data Availability Statement:** All relevant data are within the paper and its Supporting Information files.

**Funding:** The study was supported by the Ministry of Health, Czech Republic - conceptual development of research organization, Motol University Hospital, Prague, Czech Republic 00064203. The funders had no role in study design, data collection and analysis, decision to publish, or preparation of the manuscript.

**Competing interests:** The authors have declared that no competing interests exist.

associations between temperature perception test and age and BMI were proved as well. Additionally, a decrease of total protein in a patient´s serum below the physiological cut-off in the 6 months prior to measurement was associated with a pathological result of a Sudoscan. Cardiovascular autonomic neuropathy based on Ewing´s battery tests was present in 56.7% of patients, no statistically significant risk factors were found. Our peripheral neuropathy questionnaire correlated with the results of the Sudoscan test and some tests of the clinical examination of peripheral sensor-motor nervous function (discriminatory contact perception test, temperature perception test).

## Conclusions

This study demonstrated a relatively high prevalence of peripheral (especially autonomic) neuropathy and verified some risk factors for the development of peripheral somatic neuropathy in asymptomatic patients with severe form of CD on anti-TNFα biological therapy.

## Introduction

Crohn's disease (CD), which belongs to the inflammatory bowel disease, is a chronic inflammatory disease characterized by segmentar transmural inflammatory affection of the gastrointestinal tract [1]. CD can affect any part of the gastrointestinal tract, but the ileum and proximal colon are most commonly involved [2]. CD can also be associated with a wide range of extraintestinal manifestations (EIMs) such as disorders of the joints, skin, bones, eyes, hepatobiliary and other organ systems. EIMs are reported in up to 35% of patients with CD while the frequency of EIM increases with the duration of the disease and can precede the diagnosis of CD [3–5]. EIMs usually reflect intestinal disease activity with a few exceptions (uveitis, primary sclerosing cholangitis, pyoderma gangrenosum and ankylosing spondylitis). All EIMs can significantly negatively affect a patient´s quality of life and some can even significantly affect morbidity and mortality [4].

Neurological disorders are one group of CD EIMs which can affect both the peripheral and central nervous system. These neurological affections can be classified as peripheral neuropathy, demyelinating disorders, cerebrovascular disorders and medication-induced neurological manifestations. The prevalence of neurological EIMs in IBD is described as 3–39% [4, 6, 7]. The exact pathogenesis of neurological EIMs has not yet been clarified.

Peripheral neuropathy is one the main type of neurological EIM in CD and the following factors can play a causal role in its development: chronic inflammatory activity of the underlying intestinal disease, macro/micronutrient absorption disorders, intercurrent viral infections or adverse effects of CD therapy (including metronidazole, sulfasalazine or anti-TNFα biological therapy) [4, 8]. Real prevalence of peripheral neuropathy in IBD remains indeterminate and reaches up to 40% according to available studies [7, 9–11].

The autonomic nervous system (ANS) plays an essential role in the secretion, motility and mucosal immunity of the gastrointestinal tract. Rising evidence suggests that CD, within the framework of systemic inflammation, can also cause dysfunction of ANS [12, 13]. Some data also suggest that ANS dysfunction can play an essential role in the pathogenesis of IBD [14]. Based on studies of heart rate variability used routinely in evaluation of ANS dysfunction in IBD patients, ANS dysfunction could be present in approximately 5% of IBD patients [15]. However, available data are strongly inconsistent in their conclusions—normal, decreased or

also increased ANS activity are described [16]. Yet, the essential fact seems to be that subclinical affection of ANS in IBD patients could be a risk factor for increased morbidity and mortality [17].

The primary outcome of this study was to demonstrate the presence of peripheral (somatic and autonomous) neuropathy in neurologically asymptomatic (no reported difficulties in a possible association with the involvement of the peripheral nervous system until now) patients with severe form of CD on anti-TNFα biological therapy. The secondary outcome was to find risk factors for the development of peripheral neuropathy and its early predictors.

## Materials and methods

### Patients

This study was conducted at the Gastroenterology Department of Motol University Hospital in Prague, Czech Republic with outpatients > 18 years of age. Patients with CD were eligible to participate in the study if they used a medication with biological therapy from the anti-TNFα group (infliximab, adalimumab).

In order to ensure that results were not affected by potentially confounding factors, exclusion criteria were as follows: patients diagnosed with CD less than 2 years before entering the study, short-term therapy with biological therapy anti-TNFα (less than 6 months), active form of CD or severe complication of CD, psychiatric diseases, current (in last 6 months) / long-term (cumulatively longer than 1 year) metronidazole medication, other diseases associated with peripheral nervous system affection (diabetes mellitus, thyroid diseases, alcoholism, amyloidosis, severe nephropathy, severe cardiovascular diseases) and short bowel syndrome.

Concomitant therapy of CD was not identified as a contraindication. During the study, patients received the usual care from their gastroenterologists, which was not modified because of the study.

### Ethical statement

The study protocol was approved by the ethics committee of the Second Faculty of Medicine, Charles University in Prague, Czech Republic. Each patient provided written informed consent to this study prior to being accepted. The first patient was recruited to the study on 7 July 2021 and the last one on 22 March 2022. All data were processed in accordance with patient data protection.

### Methods

Patients fulfilling the inclusion criteria (men and women) were asked to answer some basic questions focusing on comorbidities, pharmacotherapy, details of their disease and therapy in order to collect information on baseline demographics and characteristics at the beginning. Furthermore, additional information about patient treatment in detail was collected from the available documentation (dose and interval of biological therapy, total length of administration of biological therapy, last drug level and presence of antibody against biological therapy, presence of fistules and their types, extraintestinal manifestations and others). CDAI was counted in all patients.

Blood samples were focused on: basic blood parameters (blood count, electrolytes, liver tests, albumin, total protein, CRP, blood lipids), parameters to exclusion pathologies possibly affecting measurement results (blood glucose and glycated hemoglobin, thyroid parameters, urea and creatinine), micronutrients, vitamins and trace elements, and basic immunological panel including serology of celiac disease.

Initially, a general neurological examination was also performed by an experienced neurologist. Furthermore, an ankle-brachial index (a quick and noninvasive test to detect asymptomatic atherosclerosis, pathological at < 0.9) was also performed to exclude pathologies possibly affecting measurement results.

Patients were also asked to complete three questionnaires:

1. The Short Inflammatory Bowel Disease Questionnaire (SIBDQ): This questionnaire is focused on measuring the health-related quality of life for Crohn's disease and ulcerative colitis. The questionnaire consists of 10 questions which reflects the physical status, social status, emotional status and general well-being of a patient. Each question has a 7-point scale (1–7 points). The absolute SIBDQ score ranges from 10 (poor health-related quality of life) to 70 (optimal health-related quality of life).

2. The sensor-motor neuropathy questionnaire (Sup.information no.1): The questionnaire consists of 12 questions focused on the presence of sensor-motor neuropathy. The sum of the answers (YES—1 point, NO—2 points) to all 12 questions is subsequently divided by 12 and if the resulting score is lower than 1.5, a suspicion for the presence of sensor-motor neuropathy is present. This questionnaire is particularly and widely used for diagnosis of diabetic peripheral neuropathy and is recommended in the guidelines of the Czech Diabetological Society for diabetic neuropathy.

3. The autonomic neuropathy questionnaire (Sup.information no.2): The questionnaire consists of 12 questions divided into 4 groups of autonomic neuropathy symptoms (cardiovascular, gastrointestinal, urogenital and skin groups). The evaluation is performed individually—any positive answer to a question is evaluated separately. In our group of patients, this questionnaire was evaluated as positive if symptoms of 2 or more groups of symptoms of autonomic neuropathy were present. This questionnaire is particularly and widely used for diagnosis of diabetic autonomic neuropathy and is recommended in the guidelines of the Czech Diabetological Society for diabetic neuropathy.

Clinical examination of peripheral sensor-motor nervous function was performed via these tests:

1. Discriminatory contact perception: This test is performed by a special 10g microfilament placed in 6 different points on both feet (scale 0–6) and focused on surface perception. Together, 6 points are investigated (3 points from the plantary side and 3 points from the dorsal side of the foot). Each point is tested repeatedly, with at least once blindly. A positive (patological) finding for neuropathy is when the patient incorrectly responds at least twice in one test point.

2. Vibration perception: This test is performed on both feet by a tuning fork (scale 0–8). This test is performed using a special graduated tuning fork (128 Hz) on the dorsal side of 1. metatarsus (scale 0–8) and focused on deep perception. It takes into account the average of 3 consecutive measurements for each leg separately. The tuner is graduated into 8 degrees. Abnormal findings in the age group up to 50 years is less than or equal to 5 and over 50 years of age lower than or equal to 3.

3. Temperature perception on both lower extremities: This test is focused on thermal sensitivity examination and is performed on both feet via a test tube with cold water (temperature 5–25˚C) or hot water (40–60˚C) repeatedly in at least 2 places on each foot. A positive (patological) finding is when the patient incorrectly responds at least twice in one test point.

Autonomic neuropathy examination: The examination is performed by evaluating the cardiovascular autonomic nerve function using a telemetry device for computer processing of heart rate variability (DiANS PF8). The standard conditions of the investigation were strictly observed. The examination was performed on an empty stomach with some regimen restrictions (no smoking, coffee or alcohol) 24 hours before the test. Medication that could interfere with measurement (anticholinergic drugs, sympathomimetics, parasympathomimetics, antihypertensives, cardiotonics or corticosteroids) were discontinued at least 24 hours before the examination.

The following was evaluated by the DiANS PF8 device:

a. Time analysis of heart rate variability (Ewing's battery of tests) - 1. deep breathing test (ratio of average heart rate in inspiration and expiration = I / E ratio); 2. Valsalva maneuver (ratio of the value of the longest R-R interval after expiration to the value of the shortest R-R interval during expiration against resistance, 40 mmHg = Valsalva ratio = VR); 3. orthostatic test (ratio of the longest to the shortest R-R interval after verticalization = RRmax / RRmin); 4. blood pressure response to verticalisation (pathological with a decrease in systolic blood pressure by more than 30 mmHg)

- The severity of cardiac autonomic neuropathy (CAN) was assessed as follows:

  ➢ borderline CAN—only 1 pathology in 1–3. Ewing's battery tests

  ➢ manifest CAN—2–3 pathologies in 1–3. Ewing's battery tests

  ➢ severe CAN—3 pathologies in 1–3. Ewing's battery tests and pathology in the 4th test

b. Spectral analysis of heart rate variability from the lying-standing-lying position (low frequency power—LF power, high frequency power—HF power, total power—TP, low frequency / high frequency ratio—LF / HF ratio)

The results for individual patients were approximated to the age of the patients using normative data for the Czech population according to Vlčková [18] in the following way:

1. Deep breathing test (I/E ratio) based on age: 20–29 years $\geq$ 1.36, 30–39 years $\geq$ 1.23, 40–49 years $\geq$ 1.14, 50–59 years $\geq$ 1.10, 60–70 years $\geq$ 1.06

2. Valsalva maneuver (VR) based on age: 20–29 years $\geq$ 1.30, 30–39 years $\geq$ 1.24, 40–49 years $\geq$ 1.20, 50–59 years $\geq$ 1.15, 60–70 years $\geq$ 1.09

3. Orthostatic test (RRmax/RRmin) based on age: 20–29 years $\geq$ 1.28, 30–39 years $\geq$ 1.20, 40–49 years $\geq$ 1.20, 50–59 years $\geq$ 1.13, 60–70 years $\geq$ 1.13

Finally, a Sudoscan was performed. Sudoscan (Impeto Medical SAS) evaluates the function of sweat glands (sudomotor function) innervated by the peripheral system. It is an examination that reveals a disorder of autonomic innervation (sudomotor function) as well as an early manifestation of sensory-motor neuropathy. Sudoscan evaluates sudomotor function on the palms of both hands and the soles of both feet. An Electrochemical Skin Conductance (ESC) is calculated and automatically evaluated by software as either normal or pathological for every extremity (depending on age, gender and ethnicity).

The primary outcome of the study was to demonstrate the presence of subclinical or manifest peripheral (somatic and autonomous) neuropathy in patients with Crohn's disease on anti-TNFα biological therapy. The secondary outcome was to find risk factors for the development of peripheral neuropathy and its early predictors.

## Statistical analyses

Standard summary statistics were used to describe primary data, median was used for cardinal data. The relationships between the measured parameters were examined using correlation coefficients. Spearman correlation coefficients were used for the correlation between two continuous (taking many different values) variables, tetrachoric correlation coefficients were used for the correlation between two binary (taking values 0/1) variables, and biserial correlation coefficients were used for the correlation between binary and continuous variables. In all versions, the correlation coefficient is a number between -1 and 1, with negative values indicating an inverse relationship between the variables (up to +/-0.4 the correlation was marked as weak, up to +/-0.7 as moderate and above +/-0.7 as strong). The value $\alpha = 0.05$ was adopted as a level of statistical significance in all analyses. Analyses were performed using STATA 13 (StataCorp LP, USA, 2013).

## Results

### Patient demographics (Table 1)

In our study, thirty eligible patients (15 men and 15 women) with CD were enrolled. The median patient age was 46 (range 30–62 years) and median patient body mass index was 25 (range 17–33). In terms of other common diseases only 2 patients (6.7%) reported arterial hypertension and no other serious comorbidities (such as thyroid gland diseases, ischemic

**Table 1. Baseline patient demographics and clinical characteristics.**

| Sex | Men, number [%] | 15 [50.0] |
|---|---|---|
| | Women, number [%] | 15 [50.0] |
| **Age before therapy, median [range]** | | 46.0 [30.0–62.0] |
| **BMI, median [range]** | | 25.0 [17.0–33.0] |
| **Duration of CD—years, median [range]** | | 16.5 [3.0–38.0] |
| **Duration of BL- years, median [range]** | | 10.0 [2.0–17.0] |
| **Type of BL** | Infliximab, number [%] | 20.0 [66.7] |
| | Adalimumab, number [%] | 10.0 [33.3] |
| **Intensified regime of BL** | Infliximab, number [%] | 5.0 [16.7] |
| | Adalimumab, number [%] | 2.0 [6.7] |
| **CDAI, median [range]** | | 78.0 [0.0–298.0] |
| **Site of disease in CD patients** | Colon only, number [%] | 4 [13.3] |
| | Small intestine only, number [%] | 17 [56.7] |
| | Small intestine and colon, number [%] | 9 [30.0] |
| | Fistulas, number [%] | 17 [56.7] |
| **Extraintestinal manifestations** | Total, number [%] | 10 [33.3] |
| | Joints, number [%] | 9 [30.0] |
| | Occular, number [%] | 1 [3.3] |
| **History of surgery** | Total, number [%] | 11 [36.7] |
| | Small intestine resection, number [%] | 6 [20.0] |
| | Ileocecal resection, number [%] | 5 [16.7] |
| | Stricturoplasty, number [%] | 3 [10.0] |
| | Right-sided hemicolectomy, number [%] | 2 [6.7] |
| **Concomitant therapy** | 5-aminosalicylates, number [%] | 10 [33.3] |
| | azathioprine, number [%] | 4 [13.3] |

CD, Crohn's disease; CDAI, Crohn's disease activity index; BL, biological therapy.

diseases of lower extremities, diabetes melitus or other) were reported. Only 10 patients (33.3%) from our group were active smokers.

The median CD duration was 16.5 years (range 3–38 years). In our group, extent of disease was: only colon (4 patients, 13.3%), only small intestine (17 patients, 56.7%), both small bowel and colon (9 patients, 30.0%). Fistulas were reported in 17 patients (56.7%): perianal fistulas (14 patients, 46.7%), enteroenteral fistulas (2 patients, 6.7%), enterocutaneous fistulas (1 patient, 3.3%) and enterovesical fistula (1 patient, 3.3%). Extraintestinal manifestations of IBD were reported by 10 patients (33.3%) (joints in 9 patients and occular in 1 patient, 30.0% and 3.3% respectively). In terms of concomitant therapy, 10 patients (33.3%) were treated with 5-aminosalicylates and 4 patients (13.3%) with azathioprine. History of surgery was reported in 11 patients (36.7%) (small bowel resection in 6 patients– 20.0%, ileocecal resection in 5 patients– 16.7%, stricturoplasty in 3 patients– 10.0%, right-sided hemicolectomy in 2 patients– 6.7%). Finally, the median Crohn's Disease Activity Index (CDAI) was 78 points (range 0–298 points).

The median duration of anti-TNFα biological therapy administration was 10 years (range 2–17 years). In terms of type of biological therapy, infliximab was used by 20 patients (66.7%) and adalimumab by 10 patients (33.3%) (3 of the patients with adalimumab had previously used infliximab). 5 patients (16.7%) had an intensified regime of biological therapy with infliximab and 2 patients (6.7%) with adalimumab.

## Blood samples

Based on the results of laboratory tests, diseases that could be related to peripheral neuropathy were excluded in each patient: diabetes mellitus (normal fasting glycemia and glycated hemoglobin in all patients), nephropathy (no significant elevation of urea and/or creatinin), hepatopathy (no significant elevation of liver tests), severe malnutrition (normal albumin and total protein), severe pathologies in micronutrients and vitamins (no significant pathologies of calcium, phosphate, magnesium, vitamin B12 and folate), disease of thyroid gland (normal fT4 and TSH in all patients) and celiac disease (normal anti-transglutaminase antibodies IgA in all patients). The results of the immunological parameters were evaluated by an experienced clinical immunologist. The findings showed very weak ANA positivity at a titre of 1:80 in 23 patients. Given these low titres as well as the absence of clinical signs of systemic disease, the results were interpreted as negative.

CRP values were normal ($\leq$ 5 mg/l) in 22 patients and pathological ($>$ 5 mg/l) in 8 patients. The median CRP values in our group of patients was 2.3 mg/l. Moderate correlation of CRP values with sensor-motor neuropathy questionnaire (p = 0.0073) and vibration perception tests (p = 0.0033 for left leg, p = 0.0129 for right leg) were proved. Statistically significant association between pathological values of CRP and: peripheral sensor-motor nervous function (p = 1.0), Sudoscan results (p = 1.0), Ewing's battery tests (p = 0.2667) and spectral analysis of heart rate variability (p = 1.0) were not proved.

## Clinical examination of peripheral sensor-motor nervous function

Discriminatory contact perception tests were pathological in 8 patients (26.7%), vibration perception tests were pathological in 1 patient (3.3%) and temperature perception tests were pathological in 9 patients (30.0%). Statistically significant association between vibration perception test and age (p = 0.001), CD duration (p = 0.002), duration of anti-TNFα biological therapy administration (p<0.001), BMI (p = 0.006) and CDAI (p<0.001) were proved. Additionally, statistically significant association between temperature perception test and age (p = 0.006) and BMI (p = 0.018) were proved.

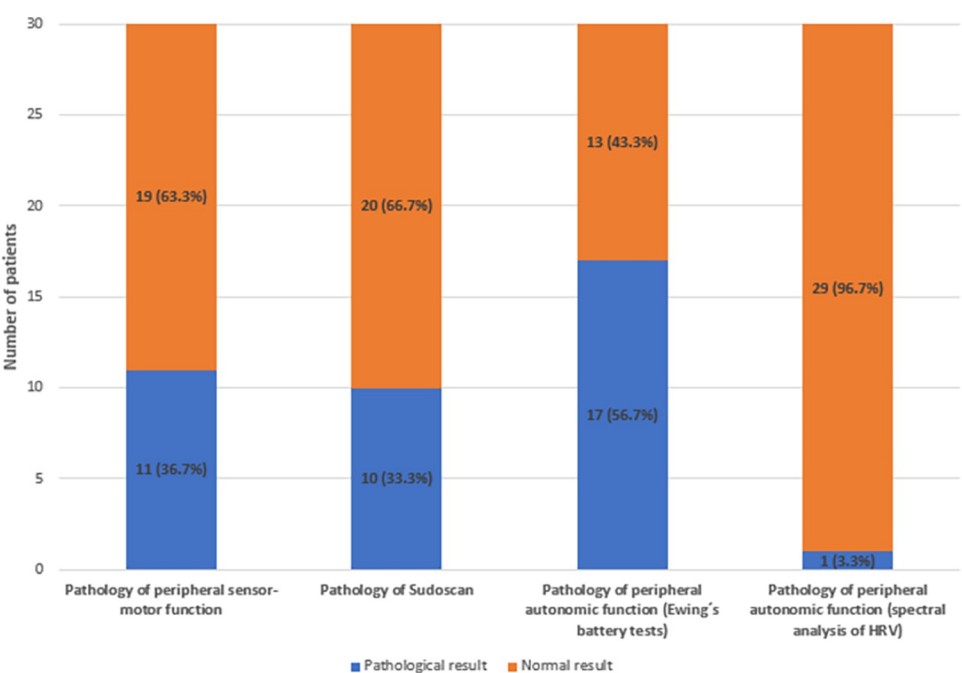

**Fig 1. Peripheral somatic and autonomic neuropathy—results of measurements.** The picture shows results of all four measured tests in our patients: clinical examination of peripheral sensor-motor nervous function (1st column), Sudoscan test (2nd column) and autonomic neuropathy examination measured by Ewing's battery of tests (3rd column) and by spectral analysis of heart rate variability (4th column). The number of patients with normal results is shown in the upper part (orange colour) and the number of patients with pathological results in the lower part (blue colour) of each column, percentages of normal and pathological values are presented in brackets for each result.

In conclusion, peripheral sensor-motor nervous function was pathological in 11 patients (36.7%) (Fig 1). BMI significantly influenced the peripheral sensor-motor nervous function (p = 0,023). On the other hand, the following factors were not found to significantly influence the presence of peripheral sensor-motor nervous function in our patients: smoking (p = 1.0), duration of CD (p = 0.599) or anti-TNFα biological therapy administration (p = 0.115), presence of other EIM (p = 1.0), abdominal surgery due to CD in personal anamnesis (p = 0.466), using mesalamine (p = 0.702), using azathioprine (p = 0.268), intensified regime of biological therapy (p = 1.0), drug level of biological therapy (p = 0.625), CDAI value (p = 0.626), and CRP value (p = 1.0).

## Sudoscan test

Ten pathological results of Sudoscan were identified in our group of patients (33.3%) (Fig 1): moderately severe dysfunction of the sudomotor function of the upper extremities in 5 patients (16.7%), moderately severe dysfunction of the sudomotor function of the lower extremities in 3 patients (10.0%), moderately severe dysfunction of the sudomotor function of the upper and lower extremities in 1 patient (3.3%), severe disorder of the sudomotor function of the upper and lower extremities in 1 patient (3.3%), severe dysfunction of the sudomotor function in upper extremities in 1 patient (3.3%).

Concomitant therapy of mesalazine significantly influenced the Sudoscan test (p = 0,045). The following factors were not found to significantly influence the presence of pathological results of Sudoscan in our patients: age (p = 0.446), BMI (p = 0.189), smoking (p = 0.420), duration of CD (p = 0.951) or anti-TNFα biological therapy administration (p = 1.0), presence

of other EIM (p = 0.231), abdominal surgery due to CD in personal anamnesis (p = 1.0), using azathioprine (p = 0.095), intensified regime of biological therapy (p = 1.0), drug level of biological therapy (p = 0.543), CDAI value (p = 0.300) and CRP value (p = 1.0). Statistically significant association between Sudoscan test results and results of peripheral sensor-motor nervous function (p = 0.247), presence of CAN (p = 1.0) and spectral analysis of heart rate variability (p = 0.333) were not proved.

## Ewing's battery of tests

Deep breathing test (I/E ratio) was pathological in 10 patients (33.3%), Valsalva maneuver test (VR) was pathological in 5 patients (16.7%), orthostatic test (RRmax/RRmin) was pathological in 9 patients (30.0%) and blood pressure response to verticalisation was pathological in 0 patients (0.0%). Based on these results CAN was present in 17 patients (56.7%) (Fig 1). The severity of CAN present in our patients was as follows (Fig 2): no CAN in 13 patients (43.3%), borderline CAN in 10 patients (33.3%), manifest CAN in 7 patients (23.3%) and severe CAN in 0 patients (0.0%).

The following factors were not found to significantly influence the presence of pathological results in Ewing's battery of tests in our patients: age (p = 0.500), BMI (p = 0.228), smoking (p = 0.056), duration of CD (p = 0.254) or anti-TNFα biological therapy administration (p = 0.592), presence of other EIM (p = 0.706), abdominal surgery due to CD in personal anamnesis (p = 0.708), using azathioprine (p = 0.290), using mesalazine (p = 0.255), intensified regime of biological therapy (p = 0.190), drug level of biological therapy (p = 1.0), CDAI value (p = 0.138) and CRP value (p = 1.0).

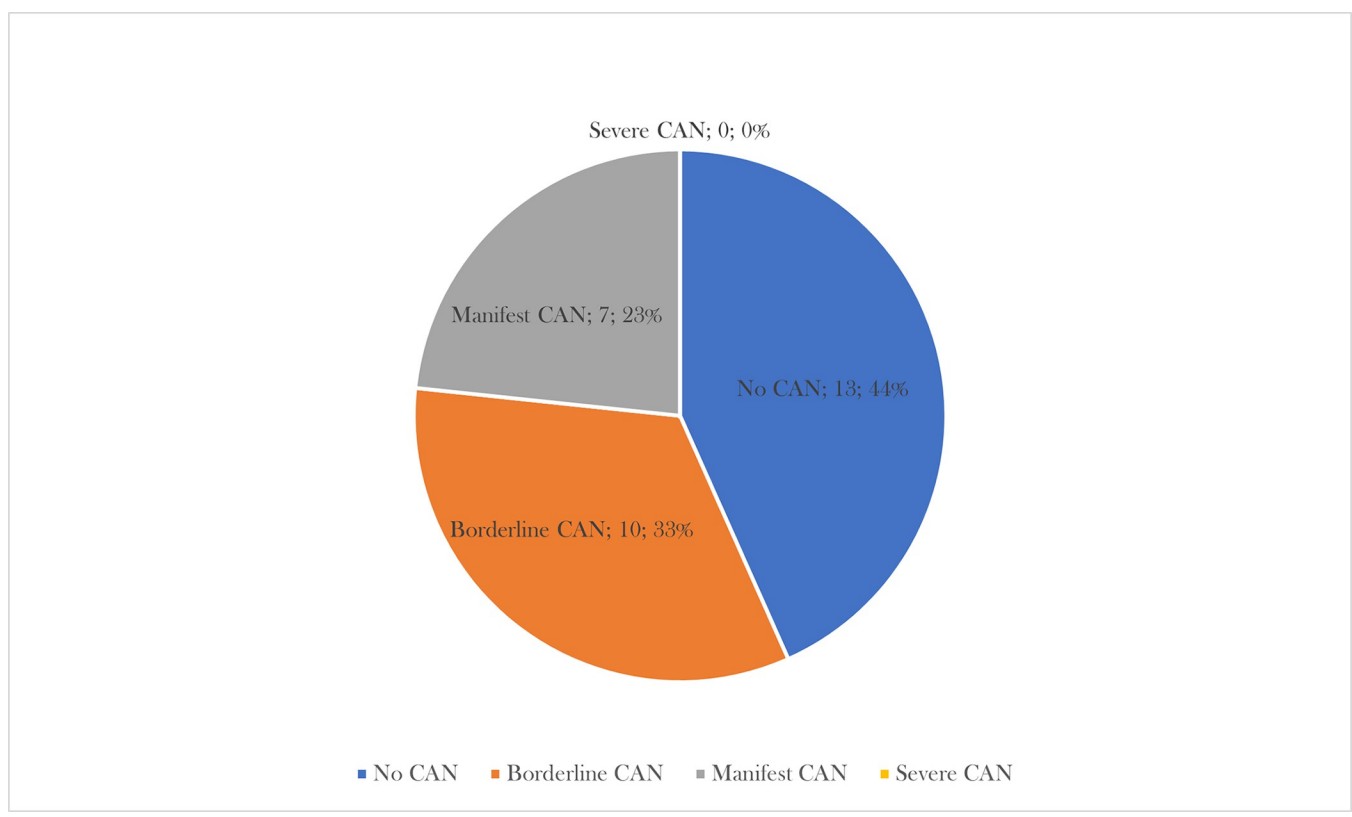

**Fig 2. CAN results and severity based on Ewing's battery of tests.** The picture shows distribution of patients according to the severity of autonomic neuropathy measured by Ewing's battery of tests.

## Spectral analysis of HRV

Only one patient had a pathological result from spectral analysis of heart rate variability in our group of patients (3.3%) (Fig 1). This patient suffered from manifest CAN according to Ewing´s battery of test and sudomotor dysfunction according to Sudoscan. Other patients had normal values of HRV spectral analysis. Statistically significant association between results of spectral analysis and Ewing's battery of tests was not proved (p = 1.0).

Duration of CD significantly influenced spectral analysis of HRV (p = 0,003). The following factors were not found to significantly influence the presence of pathological results of HRV spectral analysis in our patients: age (p = 0.167), BMI (p = 0.367), smoking (p = 1.0), duration of anti-TNFα biological therapy administration (p = 0.294), presence of other EIM (p = 0.333), abdominal surgery due to CD in personal anamnesis (p = 0.367), using azathioprine (p = 1.0), using mesalazine (p = 1.0), intensified regime of biological therapy (p = 1.0), drug level of biological therapy (p = 1.0), CDAI value (p = 1.0) and CRP value (p = 0.267).

## Questionnaires

**SIBDQ.** The median of SIBDQ results, which reflects quality of life of our patients, was 56 points (maximum = 70 points, higher score means better quality of life) with a range of 31–69 points. Moderate inverse correlation between SIBDQ and CDAI was proved (p = 0.0005) and moderate positive correlation was proved between SIBDQ and sensor-motor neuropathy questionnaire (p = 0.0001).

**Sensor-motor neuropathy questionnaire.** This questionnaire was positive in only one patient from our study group with a median result of 23 points (range 15–24 years) (Table 2). Weak asociations between sensor-motor neuropathy questionnaire and results of Sudoscan (p = 0.0039) and clinical examination of sensor-motor nervous function (discriminatory contact perception tests p = 0.0391, temperature perception tests p = 0.0215) were proved. The results of the sensor-motor neuropathy questionnaire were generally not significantly associated with the results of the clinical examination of peripheral sensor-motor nervous function (p = 1.0).

**Autonomic neuropathy questionnaire.** This questionnaire was positive in 18 patients from our study group (Table 2). The most common symptoms of autonomic neuropathy were reported: palpitations in 14 patients (46.7%), vertigo after verticalization in 14 patients

**Table 2. The sensor-motor neuropathy and autonomic neuropathy questionnaires–results.**

| | | Normal result, number [%] | Pathological result, number [%] |
|---|---|---|---|
| **The sensor-motor neuropathy questionnaire** | **Total evaluation** | **29 [96.7]** | **1 [3.3]** |
| **The autonomic neuropathy questionnaire** | **Total evaluation** | **12 [40.0]** | **18 [60.0]** |
| **→ groups of autonomic neuropathy symptoms** | **Cardiovascular** | **10 [33.3]** | **20 [66.7]** |
| | **Gastrointestinal** | **18 [60.0]** | **12 [40.0]** |
| | **Urogenital** | **16 [53.3]** | **14 [46.7]** |
| | **Skin** | **15 [50.0]** | **15 [50.0]** |
| **→ most common autonomic neuropathy symptoms** | **Palpitations** | **16 [53.3]** | **14 [46.7]** |
| | **Vertigo after verticalization** | **16 [53.3]** | **14 [46.7]** |
| | **Sweating disorders** | **17 [56.7]** | **13 [43.3]** |
| | **Feeling of fullness, hiccup** | **20 [66.7]** | **10 [33.3]** |
| | **Urinary problem** | **21 [70.0]** | **9 [30.0]** |
| | **Higher temperature intolerance** | **23 [76.7]** | **7 [23.3]** |
| | **Sexual problems** | **25 [83.3]** | **5 [16.7]** |

(46.7%), sweating disorders (sweating after eating or at night, sweating more in the upper parts of the body) in 13 patients (43.3%), feelings of fullness and hiccup in 10 patients (33.3%), urinary problems in 9 patients (30.0%), higher temperature intolerance in 7 patients (23.3%) and sexual problems in 5 patients (16.7%) (Table 2). Statistically significant association between autonomic neuropathy questionnaire and Sudoscan test results (p = 1.0), Ewing´s battery tests (p = 0.465) or spectral analysis of heart rate variability (p = 0.4) were not proved.

## Nutrition and peripheral neuropathy

A documented decrease of total protein in patient´s serum below the physiological cut-off in the 6 months prior to measurement was present in 7 (23.3%) patients. Moderate correlation of this decrease and pathological result of Sudoscan test (p = 0.026) was proved. The presence of CAN (p = 0.666) or peripheral sensor-motor nervous function (p = 1.0) was not associated with statistically significant decrease in total protein.

The decrease of total protein in the last 3 years was present in 9 (30.0%) patients. Moderate correlation of this decrease and pathological result of Sudoscan test (p = 0.030) was proved. The presence of CAN (p = 0.123) or peripheral sensor-motor nervous function (p = 0.687) was not associated with statistically significant decrease in total protein.

The documented decrease of total protein at least 12 cumulative months prior was present in 9 (30.0%) patients. This decrease of total protein was not found to significantly influence the presence of CAN based on Ewing's battery of tests (p = 0.123), Sudoscan test (p = 0.116) or peripheral sensor-motor nervous function (p = 0.225).

## Discussion

Peripheral sensor-motor neuropathy was relatively frequent in our study and, based on our measurements, present in more than one third of our patients with Crohn's disease. The actual prevalence of peripheral sensor-motor neuropathy in IBD patients is a bit controversial and, based on available studies, varies between 0% to almost 40% [7, 10, 19]. This variability is mainly due to differences in the characteristics of the studied populations and the high variability in the design of individual studies. In addition, there are only limited data of the occurrence of peripheral neuropathy in neurologically asymptomatic patients (as they were in our study) with CD. Data are also limited regarding risk factors for peripheral neuropathy. In our study, only BMI significantly influenced the peripheral sensor-motor nerve function in general, but only some of the tests (used to measure peripheral sensor-motor neuropathy) showed the following additional risk factors: age, duration of CD and anti-TNFα biological therapy and CDAI. According to a retrospective study from the Mayo clinic and consistent with our results, the incidence of peripheral neuropathy increases with disease duration (more than 3-fold increase in 10 years of disease duration) [10]. Gondim et al. reported in their study, contrary to our conclusions, the absence of correlation of peripheral neuropathy with the activity of underlying intestinal disease [20]. The relatively high prevalence of peripheral sensor-motor neuropathy in our cohort of patients could be, among others, due to the use of anti-TNFα biologics in all of our patients, which is in correlation with available studies on the role of anti-TNFα biologics in the etiopathogenesis of peripheral neuropathy in CD [21, 22]. Correlation of peripheral neuropathy with other pharmacotherapies (mesalazine, azathioprine) was not demonstrated in our study.

Based on our results, CAN was present in more than half of patients (borderline CAN in 58,8% and manifest CAN in 41,2% of patients), which is several times more frequent than the published but also very limited data [15]. However, the results of available clinical studies related to autonomic nervous system dysfunction in CD patients are not entirely consistent in

their conclusions. These results may be related to differences in the methods of evaluating autonomic neuropathy, differences in disease activity and different patient selection criteria. According to some of the available data, autonomic dysfunction in CD patients is described early in the course of the disease and thus the presence of autonomic neuropathy is not associated with the duration of CD [23, 24]. In contrast to these data, Straub et al. in their study pointed out the relative rarity of CAN in IBD patients (0% of patients with CD, in contrast to the relatively frequent presence of autonomic hyperreflexia—29% of the entire CD cohort had evidence of cardiovascular autonomic hyperreflexia), and Ganguli et al. did not demonstrate autonomic nervous system dysfunction at all in their study of CD patients compared with healthy controls [25, 26]. Dysfunction of the autonomic nervous system may be asymptomatic (even in more severe damage), but symptoms resulting from the physiological functions of the autonomic nervous system are more common: palpitations, orthostatic hypotension, dysphagia, abdominal fullness, gastrointestinal dysmotility, fecal incontinence, urinary incontinence, difficulty urinating, erectile and ejaculatory dysfunction, sweating disorders and heat intolerance. Knowledge of the potential high prevalence of autonomic neuropathy in clinical practice may have a major impact on the routine care of patients with CD. For example, symptoms related to the gastrointestinal tract could mimic the higher activity of CD with subsequent consequences. Similarly, knowledge of other symptoms of autonomic dysfunction is crucial in the quality care of these patients.

Detailed analysis of our data did not identify a statistically significant risk factor for the development of CAN. This means that in our study, the investigated risk factors for the development of autonomic neuropathy such as age, BMI, smoking, duration and also recent disease activity, specifics of biological therapy, presence of other EIMs or co-medication were not identified in causal association with the presence of CAN in our patients. Our data correlate with the very limited data on this issue in the available literature. However, Straub et al. demonstrated a correlation between systemic inflammatory response and the presence of autonomic hyperreflexia in patients with higher disease activity (higher CDAI scores) [26]. Ananthakrishnan et al. demonstrated that the administration of biologic therapy is independently associated with autonomic neuropathy, and that the increased frequency of autonomic dysfunction in patients on biologic therapy is likely associated with a more severe form of the disease and the need for a higher rate of escalation of therapy for the underlying disease [27]. These data are consistent with the relatively high frequency of autonomic neuropathy in our group of patients with severe form of CD on biologic therapy. The severity of this problem is demonstrated by a study showing that symptomatic autonomic neuropathy in IBD patients is associated with lower quality of life, more frequent psychiatric comorbidities, and higher rates of healthcare utilization [27]. Spectral analysis of HRV was pathological only in one patient in our group of patients. However autonomic neuropathy based on Ewing's battery tests was present in 17 of 30 patients (i.e., more than 50% of patients). Ewing's battery tests as a safe, noninvasive and well standardized method is based on both our and other availabe data which demonstrate it as the gold standard in clinical testing of autonomic neuropathy [28]. Based on our data, Ewing's battery tests were more sensitive in the diagnosis of autonomic neuropathy. The only patient with pathological spectral analysis of HRV also had manifest CAN according to Ewing´s battery of tests and sudomotor dysfunction according to Sudoscan. Although causality was not established, spectral analysis of HRV was useful in our study in the diagnosis of more severe forms of peripheral neuropathy.

Our study was the first one which evaluated the role of Sudoscan in the diagnosis of peripheral neuropathy in IBD patients. Sudoscan is a simple and fast method based on the mechanisms of reverse iontophoresis and chronoamperometry in the sympathetic innervation of the sweat glands of the extremities [29]. The role of sudoscan is in the diagnosis of peripheral

neuropathy, specifically its autonomic component (sudomotor function). In clinical practice, it has applications and most data in the evaluation of diabetic polyneuropathy [30]. Significant correlations of Sudoscan and sweat gland density nerve fiber density in skin biopsies [31], conventional techniques of nerve conduction studies and electromyography [32] and relationship with CAN [33, 34] were proved. Another advantage is the early detection of peripheral neuropathy when using Sudoscan [35]. Pathological Sudoscan was present in one third of our patients and we proved sudomotor dysfunction of the peripheral nervous system in these patients. Consistency in Sudoscan and CAN and measurement of peripheral sensor-motor nerve function was present in 50% and 43.3% of patients, respectively. Statistical significance in the association of these parameters was not demonstrated in our study. This discrepancy could be explained by the faster regenerative capacity of the thin nonmyelinated C-fibres responsible for sweating [36].

Relatively high frequencies of peripheral somatic and autonomic neuropathy in our study could be influenced by the spectrum of included patients. All of the patients in our study had Crohn's disease on biological therapy with TNFα inhibitors. Biological therapy was indicated in our patients if they met the indication criteria for it as per the Summary of Product Characteristics for this biological agent in the Czech Republic. This means that biological therapy with TNFα inhibitors was indicated for treatment of patients with moderate to severe active CD or fistulising active CD who had not responded despite a full and adequate course of therapy (corticosteroid or immunosuppressive therapy), who were intolerant to these types of therapies, or who had medical contraindications for these therapies. It follows from the above that the spectrum of our patients was primarily a group of patients with a severe form of CD with possible effects and consequences of this chronic inflammatory disease. Pathogenesis of Crohn's disease is associated among others with several mechanisms of immune system dysregulations. All EIMs are usually associated with a more severe form of CD [3, 4]. Extension of immune responses from the intestine and immune responses in extraintestinal sites are important pathogenic mechanisms in the development of EIMs, and immune-mediated mechanisms have also been described as crucial to the development of neurological EIMs [4, 8, 37, 38]. Despite recent low values of CDAI, which reflect only recent activity of disease in our spectrum of patients with severe form of CD, these immune mechanisms may have been accentuated in the past and now may play a role in the relatively higher frequency of positive outcome measures in our study.

Malnutrition in patients with IBD, but more often in patients with CD, is a very frequent and serious complication with possible acute and chronic consequences [39]. It is present in up to 75% of patients at the time of diagnosis [39]. The causes of malnutrition (reduced food intake, disturbances of intestinal motility, increased catabolism, reduced absorptive surface area, depressive syndrome and others) are often combined [39]. The consistency in the results of physiological/pathological values of total protein with the measurements of neuropathy in our patients was as follows: in 40% of the cases it was consistent with the results of the Ewing´s battery test, in 60% of the cases it was consistent with the results of the peripheral sensor-motor neuropathy measurements and in 76.7% of the cases it was consistent with the Sudoscan results. A statistically significant association was demonstrated only in the case of Sudoscan. The results may certainly have been influenced by the absence of available data on the severity and duration of malnutrition before and at the time of diagnosis of CD, as well as in the early stages of the disease, which were certainly present given the frequency of malnutrition in the early stages of the disease [39]. Malnutrition plays a major pathogenetic role in the development of neuropathy in patients with IBD [4]. In our opinion and according to our results, the role of malnutrition is one of the main ones in the development of neuropathy in CD patients. Early diagnosis, therapy and even better prevention of malnutrition could be a crucial step in

minimizing acute and, especially, chronic consequences (including peripheral neurological EIMs) of malnutrition. Malnutrition is not associated only with macronutrient deficiencies, but also with micronutrient and vitamin depletions. Vitamin and micronutrient deficiencies (among others) may play a major role in the etiopathogenesis of peripheral neuropathy in Crohn's disease—specifically, vitamin B12 and folate deficiency secondary to malabsorption syndrome [19]. Despite the absence of significant depletion of the micronutrients and vitamins in our group of patients at the time of measurement, it could be likely that, given the severe form of the underlying disease, deficiency was present in some kind of degree and for some time in the past and may have played a significant role in the development of peripheral neuropathy. Certainly, this issue will be the subject of further investigation.

In our study three kinds of questionnaire were use. The first, SIBDQ, proved recent relatively good quality of life of our patients, which corelates with the clinical remission of their disease. The second was the sensor-motor neuropathy questionnaire. Due to the lack of availability of a questionnaire focused on detecting peripheral neuropathy in IBD patients, a widely used questionnaire for detection of diabetic peripheral neuropathy from the guidelines of the Czech Diabetological Society was used. Correlations from this questionnaire with results from Sudoscan and some subtests from clinical examination of sensor-motor nervous function were proved. Thus, according to these data, this questionnaire seems to be a useful method to detect peripheral neuropathy in routine clinical practice. The third questionnaire was focused on detecting autonomic neuropathy. Also, here there is no specific questionnaire available for detecting autonomic neuropathy in IBD patients, and therefore the questionnaire widely used for diagnosis of diabetic autonomic neuropathy and recommended in the guidelines of the Czech Diabetological Society was used. This questionnaire was positive in 60% of our patients —in only 43.3% of cases the results were identical with the results of CAN measurement by Ewing's battery tests and in 46.7% of cases with Sudoscan. Statistically significant associations between these results were not demonstrated. The sensitivity of this questionnaire (or its future modifications) to detect autonomic neuropathy in IBD patients will have to be re-evaluated in further studies.

Limitations of the current study included:

- a relatively small number of included patients

- no control group of patients with nonrandomised and non-blinded design of our study

## Conclusion

Our study proved a relatively high prevalence of peripheral somatic neuropathy (measured by the peripheral sensor-motor nervous function test and Sudoscan) and, especially, peripheral autonomic neuropathy (measured by Ewing´s battery tests) in asymptomatic patients with severe form of CD on anti-TNFα biological therapy. Our study verified some risk factors for the development of peripheral somatic neuropathy. However, some concerns still remain in identifying risk factors for the development of peripheral autonomic neuropathy in patients with CD. Therefore, further studies are necessary to provide further data about the prevalence and, especially, the risk factors for etiopathogenesis of peripheral neuropathy.

## Supporting information

**S1 File. The sensor-motor neuropathy questionnaire.**
(DOCX)

**S2 File. The autonomic neuropathy questionnaire.**
(DOCX)

**S1 Table. Source data of this study.**
(XLSX)

## Author Contributions

**Conceptualization:** Martin Wasserbauer, Sarka Mala, Katerina Stechova, Dita Pichlerova, Barbora Kucerova, Jan Kral.

**Data curation:** Martin Wasserbauer, Jiri Drabek, Barbora Kucerova.

**Formal analysis:** Martin Wasserbauer, Sarka Mala, Katerina Stechova, Jan Stovicek, Lucia Bartuskova.

**Investigation:** Martin Wasserbauer, Stepan Hlava, Pavlina Cernikova, Jan Kral.

**Methodology:** Martin Wasserbauer, Stepan Hlava, Jan Stovicek, Jan Broz, Petra Liskova.

**Project administration:** Martin Wasserbauer, Stepan Hlava, Jan Stovicek.

**Resources:** Jan Broz.

**Software:** Pavlina Cernikova, Lucia Bartuskova.

**Supervision:** Sarka Mala, Katerina Stechova, Jan Stovicek, Jiri Drabek, Jan Broz, Petra Liskova, Radan Keil.

**Validation:** Sarka Mala, Dita Pichlerova, Petra Liskova, Lucia Bartuskova.

**Visualization:** Pavlina Cernikova, Jiri Drabek.

**Writing – original draft:** Martin Wasserbauer, Stepan Hlava.

**Writing – review & editing:** Jan Broz, Dita Pichlerova, Barbora Kucerova, Radan Keil.

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
