## [Decision Letter · Decision Letter 0]

26 Sep 2023

PONE-D-23-25133Dysfunction of peripheral somatic and autonomic nervous system in patients with severe forms of Crohn's disease on biological therapy with TNFα inhibitors – a single center study.PLOS ONE

Dear Dr. Hlava,

Thank you for submitting your manuscript to PLOS ONE. After careful consideration, we feel that it has merit but does not fully meet PLOS ONE’s publication criteria as it currently stands. Therefore, we invite you to submit a revised version of the manuscript that addresses the points raised during the review process.

We look forward to receiving your revised manuscript.

Kind regards,

Junji Xing, Ph.D.

Academic Editor

PLOS ONE

Journal Requirements:

Reviewers' comments:

Reviewer's Responses to Questions

**Comments to the Author**

1. Is the manuscript technically sound, and do the data support the conclusions?

Reviewer #1: Yes

Reviewer #2: Yes

2. Has the statistical analysis been performed appropriately and rigorously? 

Reviewer #1: Yes

Reviewer #2: Yes

3. Have the authors made all data underlying the findings in their manuscript fully available?

Reviewer #1: Yes

Reviewer #2: Yes

4. Is the manuscript presented in an intelligible fashion and written in standard English?

Reviewer #1: Yes

Reviewer #2: Yes

5. Review Comments to the Author

Reviewer #1: This manuscript investigated neurological issues in severe Crohn's disease patients on anti-TNFα therapy. It found a high prevalence of peripheral neuropathy (36.7%) and cardiovascular autonomic neuropathy (56.7%). Risk factors included age, therapy duration, BMI, and disease activity. The study also highlighted the importance of neurological assessments in these patients. However，in the manuscript, there are areas where refinement and improvement are still necessary.

1. The figure legend needs to be supplemented in detail, for example, the numbers on the bars in Figure 1 represent the number of individuals, and it would be best if the statistical differences (p-values) could be reflected on the figure.

2. Please present the results of the two questionnaires in the form of tables or figures

3. Many of the data descriptions in the manuscript cannot be directly obtained from the provided tables and figures. Please provide additional information, for example, but not limited to the results of blood samples and the data from lines 296 to 302.

Reviewer #2: The submitted manuscript has done an excellent job of demonstrating a high prevalence of peripheral neuropathy in patients with severe Crohn's disease on anti-TNF alpha therapy. Specific risk factors have also been presented which are responsible for the development of this neuropathy in patients.

Overall the manuscript is very detailed, well explained and the data has been presented in a clear manner. However, there are certain minor errors which need to be addressed :-

1. Page 4, Line 58 : The spelling of "inflamatory" is incorrect.

2. Page 6, Line 101 : It is suggested to replace the word "realized" with "conducted".

3. Page 6, Line 107 : Please keep the format of anti-TNF alpha consistent.

4. Page 7, Line 131 : What are "ionts"?

5. Page 7, Line 135 : "Coeliac Disease" is misspelled.

6. Page 14, Line 276 : The spelling of "melittus" is incorrect.

6. PLOS authors have the option to publish the peer review history of their article (what does this mean?). If published, this will include your full peer review and any attached files.

Reviewer #1: No

Reviewer #2: No

---

## [Author Response · Author response to Decision Letter 0]

6 Oct 2023

Dear reviewers and editors,

 First of all, I would like to thank you very much for your beneficial recommendations and interesting comments to our manuscript. All your comments were reflected during revision of our manuscript.

Below I would like to present you these manuscript changes in details:

Journal Requirements:

- Manuscript was checked and corrected based on journal requirements according to sent pdf documents.

- Grant information will be checked during resubmitting of the revised version of our manuscript.

- Based on the internal requirements of our faculty (for reasons of financing the publication process), we would like to ask you to cite our funding statement (Supported by the Ministry of Health, Czech Republic - conceptual development of research organization, Motol University Hospital, Prague, Czech Republic 00064203.) somewhere as part of our manuscript.

- All references were checked (if you have a problem with some specific reference, please notify us of this and we will certainly reflect that).

Reviewer's Comments to the Author

Reviewer #1: 

This manuscript investigated neurological issues in severe Crohn's disease patients on anti-TNFα therapy. It found a high prevalence of peripheral neuropathy (36.7%) and cardiovascular autonomic neuropathy (56.7%). Risk factors included age, therapy duration, BMI, and disease activity. The study also highlighted the importance of neurological assessments in these patients. However，in the manuscript, there are areas where refinement and improvement are still necessary. 

Comments:

1. The figure legend needs to be supplemented in detail, for example, the numbers on the bars in Figure 1 represent the number of individuals, and it would be best if the statistical differences (p-values) could be reflected on the figure.

- Figure legends were implemented in details according to your beneficial recommendation.

- Figure 1 summarizes the results of the four main types of measurements in our study; correlations were not included in the figure mainly due to the small number of patients in our study. The graph in Figure 1 was modified in response to your comments for better reproducibility of results - data labels of results, vertical axis, percentages of normal and pathological measurement results.

2. Please present the results of the two questionnaires in the form of tables or figures

- Manuscript was edited based on your requirements, the table with the results of both questionnaires was added to the manuscript.

3. Many of the data descriptions in the manuscript cannot be directly obtained from the provided tables and figures. Please provide additional information, for example, but not limited to the results of blood samples and the data from lines 296 to 302. 

- Manuscript was edited based on your requirements, table with source data of our study was added as the separate additional file with our resubmitted manuscript.

Reviewer #2: 

The submitted manuscript has done an excellent job of demonstrating a high prevalence of peripheral neuropathy in patients with severe Crohn's disease on anti-TNF alpha therapy. Specific risk factors have also been presented which are responsible for the development of this neuropathy in patients. Overall the manuscript is very detailed, well explained and the data has been presented in a clear manner. However, there are certain minor errors which need to be addressed:

1. Page 4, Line 58 : The spelling of "inflamatory" is incorrect.

- Manuscript was corrected according to your recommendation.

2. Page 6, Line 101 : It is suggested to replace the word "realized" with "conducted".

- Manuscript was corrected according to your recommendation.

3. Page 6, Line 107 : Please keep the format of anti-TNF alpha consistent.

- Manuscript was corrected according to your recommendation.

4. Page 7, Line 131 : What are "ionts"?

- Ionts were incorrectly mentioned in the text instead of electrolytes, manuscript was corrected.

5. Page 7, Line 135 : "Coeliac Disease" is misspelled.

- Manuscript was corrected according to your recommendation.

6. Page 14, Line 276 : The spelling of "melittus" is incorrect.

- Manuscript was corrected according to your recommendation.

---

## [Decision Letter · Decision Letter 1]

2 Nov 2023

Dysfunction of peripheral somatic and autonomic nervous system in patients with severe forms of Crohn's disease on biological therapy with TNFα inhibitors – a single center study.

PONE-D-23-25133R1

Dear Dr. Hlava,

We’re pleased to inform you that your manuscript has been judged scientifically suitable for publication and will be formally accepted for publication once it meets all outstanding technical requirements.

Kind regards,

Junji Xing, Ph.D.

Academic Editor

PLOS ONE

Additional Editor Comments (optional):

Reviewers' comments:

Reviewer's Responses to Questions

**Comments to the Author**

1. If the authors have adequately addressed your comments raised in a previous round of review and you feel that this manuscript is now acceptable for publication, you may indicate that here to bypass the “Comments to the Author” section, enter your conflict of interest statement in the “Confidential to Editor” section, and submit your "Accept" recommendation.

Reviewer #1: All comments have been addressed

Reviewer #2: All comments have been addressed

2. Is the manuscript technically sound, and do the data support the conclusions?

Reviewer #1: Yes

Reviewer #2: Yes

3. Has the statistical analysis been performed appropriately and rigorously? 

Reviewer #1: Yes

Reviewer #2: Yes

4. Have the authors made all data underlying the findings in their manuscript fully available?

Reviewer #1: Yes

Reviewer #2: Yes

5. Is the manuscript presented in an intelligible fashion and written in standard English?

Reviewer #1: Yes

Reviewer #2: Yes

6. Review Comments to the Author

Reviewer #1: The author's prompt and diligent response to the reviewers' comments, coupled with their thorough revisions, has significantly enhanced both the professionalism and readability of the manuscript. I wholeheartedly support its publication.

Reviewer #2: The authors have done a good job in addressing the issues raised by the reviewers-they have been clear in their explanation.

7. PLOS authors have the option to publish the peer review history of their article (what does this mean?). If published, this will include your full peer review and any attached files.

Reviewer #1: No

Reviewer #2: No

---

## [Editor Report · Acceptance letter]

7 Nov 2023

PONE-D-23-25133R1 

Dysfunction of peripheral somatic and autonomic nervous system in patients with severe forms of Crohn's disease on biological therapy with TNFα inhibitors – a single center study. 

Dear Dr. Hlava:

I'm pleased to inform you that your manuscript has been deemed suitable for publication in PLOS ONE. Congratulations! Your manuscript is now with our production department. 

Kind regards, 

on behalf of

Dr. Junji Xing 

Academic Editor

PLOS ONE